# A new tool for converting food frequency questionnaire data into nutrient and food group values: FETA research methods and availability

Angela A Mulligan,[1] Robert N Luben,[1] Amit Bhaniani,[1] David J Parry-Smith,[1] Laura O'Connor,[2] Anthony P Khawaja,[1] Nita G Forouhi,[2] Kay-Tee Khaw[1,3]

▶ Prepublication history and Additional material is available. To view please visit the journal (http://dx.doi.org/10.1136/bmjopen-2013-004503).

NGF and K-TK contributed equally.

For numbered affiliations see end of article.

Correspondence to
A Mulligan;
angela.mulligan@phpc.cam.ac.uk

## ABSTRACT

**Objectives:** To describe the research methods for the development of a new open source, cross-platform tool which processes data from the European Prospective Investigation into Cancer and Nutrition Norfolk Food Frequency Questionnaire (EPIC-Norfolk FFQ). A further aim was to compare nutrient and food group values derived from the current tool (FETA, FFQ EPIC Tool for Analysis) with the previously validated but less accessible tool, CAFÉ (Compositional Analyses from Frequency Estimates). The effect of text matching on intake data was also investigated.

**Design:** Cross-sectional analysis of a prospective cohort study—EPIC-Norfolk.

**Setting:** East England population (city of Norwich and its surrounding small towns and rural areas).

**Participants:** Complete FFQ data from 11 250 men and 13 602 women (mean age 59 years; range 40–79 years).

**Outcome measures:** Nutrient and food group intakes derived from FETA and CAFÉ analyses of EPIC-Norfolk FFQ data.

**Results:** Nutrient outputs from FETA and CAFÉ were similar; mean (SD) energy intake from FETA was 9222 kJ (2633) in men, 8113 kJ (2296) in women, compared with CAFÉ intakes of 9175 kJ (2630) in men, 8091 kJ (2298) in women. The majority of differences resulted in one or less quintile change (98.7%). Only mean daily fruit and vegetable food group intakes were higher in women than in men (278 vs 212 and 284 vs 255 g, respectively). Quintile changes were evident for all nutrients, with the exception of alcohol, when text matching was not executed; however, only the cereals food group was affected.

**Conclusions:** FETA produces similar nutrient and food group values to the previously validated CAFÉ but has the advantages of being open source, cross-platform and complete with a data-entry form directly compatible with the software. The tool will facilitate research using the EPIC-Norfolk FFQ, and can be customised for different study populations.

## INTRODUCTION

Food Frequency Questionnaires (FFQs) are commonly used in epidemiological studies to

## Strengths and limitations of this study

- FETA (Food Frequency Questionnaire European Prospective Investigation into Cancer and Nutrition Tool for Analysis) has been tested using a large study sample of food intake data.
- No independent reference method used in the comparisons of FETA and CAFÉ (Compositional Analyses from Frequency Estimates) nutrient intake data although the CAFÉ system has been previously validated.
- Ability to modify the underlying data files in FETA to customise it for different study populations.

assess the dietary intake of large populations. Their popularity derives from ease of administration, ability to assess dietary intake over a defined period of time and low costs.[1] The European Prospective Investigation into Cancer and Nutrition (EPIC)-Norfolk FFQ is semiquantitative and designed to record the average intake of foods during the previous year. The principles involved in data collection and processing of the EPIC-Norfolk FFQ and the development of the structure and content of the CAFÉ (Compositional Analyses from Frequency Estimates) programme for calculating nutrient intakes have been published previously.[2] The EPIC-Norfolk FFQ has been extensively validated and has been widely used.[3–5] However, the programmes used to process these FFQs, including CAFÉ, have not been easily accessible to end-users.

Our objective was to develop a new, open source, cross-platform processing tool (FETA—FFQ EPIC Tool for Analysis) based on and building on the earlier system, CAFÉ.[2] The aim of this report was to describe the research methods of the development of FETA, and to compare nutrient output from the FETA and CAFÉ

programmes. Food group intake data from FETA has also been described as having the effect of free text matching on nutrient and food group intake data. Free text matching refers to the assigning of an appropriate food code to handwritten text in the FFQ and will be further described in the methods section.

## METHODS
### EPIC-FFQ design
The questionnaire consists of two parts. Part 1 consists of a food list of 130 lines; each line has a portion size attached to it: medium serving, standard unit or household measure. Study participants were requested to select an appropriate frequency of consumption for each line, from the nine frequency categories. As an example, figure 1 illustrates the sections relating to bread, savoury biscuits and breakfast cereals. A pdf copy of the EPIC-Norfolk FFQ may be downloaded from http://www.srl.cam.ac.uk/epic/epicffq/websitedocumentation.html; information on how to complete and code the FFQ is also available here. The questionnaire lines are either individual foods, combinations of individual foods or food types. The FFQ food list is based on items from an FFQ widely used within the USA,[6] [7] but modified to reflect differences in American versus UK brand names, and some further food items were added.

Part 2 contains further questions, a number of which ask for more detailed information that link back to food lines in part 1, as illustrated in figure 2. Detailed information was requested for breakfast cereals and fats as these are nutritionally important foods in the UK diet.

### Data collection
The EPIC-Norfolk FFQ was posted to 25 639 participants in the EPIC-Norfolk cohort study.[8] The participants were aged 40–79 years, and the questionnaire was completed between 1993 and 1997. The study was approved by the Norfolk Local Research Ethics Committee, adhered to the Declaration of Helsinki, and all participants gave written informed consent. The FFQ was returned at a health examination, where it was checked and completed, if required, by trained nursing staff. In total, 25 351 (99%) participants returned the completed questionnaire.

### Comparison of FETA and CAFÉ programmes
FETA uses a comma-separated values input file. Part 1 is coded as numeric values and part 2 is coded as numeric values and food codes, using the flowcharts and look-up lists provided (http://www.srl.cam.ac.uk/epic/epicffq/). We have also created a Microsoft Access form-based entry tool to facilitate FFQ data entry, based on the EPIC-Norfolk FFQ. The tool exports data in a format directly compatible with FETA. The FETA software was written in C and C++ languages, enabling faster processing times than SAS and the C/C++ software can also be used from the command line. The step-based graphical wizard for running FETA was written in Perl. Whereas in the CAFÉ programme, an Oracle-based entry system (Oracle Corporation, Redwood Shores, California, USA) was created to enter part 1 frequency data as numeric codes and part 2 data as numeric codes and free text. CAFÉ was written using SAS (SAS Software, V.8 of the SAS System for UNIX, SAS Institute Inc, Cary, North Carolina, USA) and links to tables in an Oracle relational database.

### Part 1: Data entry
Data were manually entered into a spreadsheet as numeric codes, using '1' for 'never or less than once a month', to '9' for '6+ times per day'. A code of '−9' was used to mark data where a frequency was not recorded. Where two frequencies were provided for a line, this was coded as '−4' and treated by CAFÉ and FETA programmes as missing data. However, in FETA, both frequencies may now be entered, separated by a semicolon, for example, '2;3', and FETA will process the first value.

### Part 2: Assigning of food codes to ticked boxes and free text
Part 2 contains handwritten text for milk, breakfast cereals and cooking fats (see figure 2, questions 3, 5, 6 and 7, respectively), which needs to be matched to the most appropriate food code in order to obtain nutrient data; this process is known as free text matching. The data in part 2 were coded using reference lists of food

**Figure 1** Part 1 (main part) of the EPIC-Norfolk FFQ, illustrating bread, savoury biscuits and breakfast cereals.

| FOODS AND AMOUNTS | AVERAGE USE LAST YEAR | | | | | | | | |
|---|---|---|---|---|---|---|---|---|---|
| **BREAD AND SAVOURY BISCUITS** (one slice or biscuit) | Never or less than once/month | 1-3 per month | Once a week | 2-4 per week | 5-6 per week | Once a day | 2-3 per day | 4-5 per day | 6+ per day |
| White bread and rolls | | | | | | ✓ | | | |
| Brown bread and rolls | | | | ✓ | | | | | |
| Wholemeal bread and rolls | ✓ | | | | | | | | |
| Cream crackers, cheese biscuits | | | ✓ | | | | | | |
| Crispbread, eg. Ryvita | | | ✓ | | | | | | |
| **CEREALS** (one bowl) | | | | | | | | | |
| Porridge, Readybrek | | | | ✓ | | | | | |
| Breakfast cereal such as cornflakes, muesli etc. | | | | | ✓ | | | | |

**Figure 2** Questions from part 2 of the EPIC-Norfolk FFQ, used by FETA.

codes for varieties of milk, breakfast cereal and cooking fat. Where there is no clear match, it is suggested that a researcher consults the ingredients and nutrient information of the commercial item and compares this information with the nutrient profile of similar items from the reference lists. These reference lists and figures relating to food codes that may be assigned to appropriate ticked boxes may be found at http://www.srl.cam.ac.uk/epic/epicffq/websitedocumentation.html

Differences between FETA versus CAFÉ processing may also be found at http://www.srl.cam.ac.uk/epic/epicffq/websitedocumentation.html; these differences relate to breakfast cereals, frying and baking fats, the outcome of selecting the 'None' or 'No' box, and default milk, cereal and fat codes.

## Databases

Each line in part 1 of the FFQ is mapped to up to six food codes. Decisions regarding which food codes to use were based on data from UK government surveys and other UK population data.[7 9 10] These decisions were based on data for individuals aged 40–74 years.[7] Data for portion weights were sourced from UK population data and weighed records in 40–74-year-old study participants.[7 11]

The EPIC-Norfolk FFQ uses 290 foods from the UK food composition database, McCance and Widdowson's 'The Composition of Foods' (5th edition) and its associated supplements.[12–21] A number of new food items were added to the EPIC-Norfolk FFQ food list, which are used in the FETA and CAFÉ programmes. These include low-calorie/diet fizzy drinks and crunchy oat cereal, as well as modified home-baked and fried foods (without their fat), to enable an individual's fat type, as recorded in part 2 of the FFQ, to be incorporated. However, the nutrient data of six of the nine new foods used in the CAFÉ programme were modified in FETA. These foods include crunchy oat cereal, milk non-specific, low-calorie/diet fizzy drinks, solid vegetable oil, Crisp 'n Dry (solid fat), and oil and fat non-specific. Modifications to the nutrient data were made to ensure a more accurate nutrient profile and/or to better reflect the foods consumed, in the case of non-specific items, such as milk and oil/fat; these changes relate to nutrient/food data at the time of FFQ completion.

## Identification of outliers

Outliers were defined as detailed previously.[2] In brief, the ratio of energy intake (EI) to basal metabolic rate (BMR) was calculated, where BMR was calculated using sex-specific Schofield equations, which included age and body weight.[22] Individuals in the top and bottom 0.5% of EI:BMR ratio were identified and excluded, as were

individuals with FFQs containing 10 or more missing lines of data in part 1 of the FFQ.

## Nutrient and food group outputs

FETA produces four nutrient output formats and a sample of each of these can be viewed at http://www.srl.cam.ac.uk/epic/epicffq/websitedocumentation.html

Output 1 contains average daily nutrient and food group intakes for an individual from all FFQ foods consumed, in wide format, suitable for import into a spreadsheet or statistical package. Intake data for 46 nutrients are provided as well as data for 14 basic food groups; however, only a selection of these nutrients is shown in this report. Output 2 contains the same nutrient intake data as output 1, but in long format, which is mostly suitable for programmers. Output 3 contains average daily nutrient and food group intakes (and amount of food consumed) for an individual for each FFQ line; this output file will be very large and is mostly suitable for programmers. The most detailed output (output 4) contains average daily nutrient and food group intakes, in addition to the amount of food consumed for an individual, for each food code, for each FFQ line (meal_id). An online description of each meal_id and nutrient code, including units of measurement, can be found in the data entry template. This output will also be very large and is mostly suitable for programmers.

A log file is created along with each output file, which records the processing of the data and provides useful error information (see online supplementary appendix 1 for log file of output 1). In these files, notes (general process information) and error messages are recorded, with a date and time stamp. The log files make it possible to calculate the number of missing frequencies based on part 1 (main grid) of the FFQ in order to exclude individuals with 10 or more missing ticks. The log files also record situations where a food code does not have any nutrient data attached to it.

## Statistical analyses

The data were analysed using STATA V.10 (STATA Corp, Texas, USA). Intake data were described using mean, SD, median, minimum and maximum for FETA and CAFÉ programme outputs, stratified by sex. The nutrients selected for comparison are those described in the original CAFÉ paper. Where data on quintile changes are shown, cut-off points were calculated using CAFÉ nutrient data in order to compare quintile shift between FETA and CAFÉ output data.

## RESULTS

We received FFQs from 25 351 participants (11 451 men and 13 900 women), with a mean age of 59 years. From this set, 249 FFQs (90 men and 159 women) containing 10 or more missing lines of data in part 1 of the FFQ were excluded, followed by a further exclusion of 250 FFQs (111 men and 139 women) from the top and

bottom 0.5% of EI:BMR. This resulted in the final analytical dataset of 24 852 participants (11 250 men and 13 602 women).

## Nutrient intake data from FETA and CAFÉ programmes

Table 1 shows the average daily intake data for a number of selected nutrients for 11 250 men. The data were similar for most nutrients across the two programmes. The nutrients which had the highest percentage of quintile change ($\geq 10\%$) were monounsaturated fat, saturated fat, iron, vitamin D and vitamin E. However, only 1.3% of the men changed more than one quintile, for two of these five nutrients. The nutrients which had the lowest percentage of quintile changes were alcohol, calcium and carotene, with less than 3% change (table 1).

Table 2 shows average daily intake data for the selected nutrients for 13 602 women, from FETA and CAFÉ programmes. There were similar quintile changes observed in women to those found in men for the selected nutrients; 4 of the 19 nutrients had a quintile change of greater than 10%: polyunsaturated fat, saturated fat, iron and vitamin E. However, the number of women who shifted more than one quintile was generally lower than the number observed in men. The nutrients which had the greatest percentage of women who changed more than one quintile were vitamins D and E, with 0.7% and 0.9%, respectively.

Detailed (output 4) nutrient intake data at the individual level obtained from the two programmes were compared for approximately half of the participants (n=12 500; data not shown). All differences (>0.1%) found were investigated and explanations for these differences are considered in the discussion.

## Food group intake data from FETA

Average daily intakes for men and women of the 14 food groups readily available from FETA are shown in table 3. Mean daily intakes of six of the food groups were higher in men than in women: alcohol, cereals, fats, meat, potatoes and sugars. However, women had higher intakes of fruit (278 vs 212 g) and vegetables (284 vs 255 g). Mean daily intakes of eggs, fish, milk, non-alcoholic beverages, nuts and seeds, and soups and sauces were similar in men and women.

## The effect of text matching in FETA

Tables 4 and 5 illustrate the variation in nutrient and food group intake data obtained in a random subset of 1159 men and 1340 women, respectively, depending on whether text matching of milks, breakfast cereals, and baking and frying fats was applied. In general, mean nutrient intakes were higher when text matching was carried out. In men, (table 4), quintile changes (>15%) were most evident in the following nutrients: Englyst fibre, polyunsaturated fat, folate, vitamin D and vitamin E. The food group 'cereals and cereal products' was the only 1 of the 14 groups where there was a difference, with 31 men moving one quintile.

**Table 1** Average daily nutrient intakes for men (N=11 250) participating in the EPIC-Norfolk study, from the FETA and CAFÉ programmes, after the exclusion of outliers, with numbers and percentages of men who moved quintile

| Nutrient | FETA programme | | | | | CAFÉ programme | | | | | Quintile change | | Quintile change >1 | |
|---|---|---|---|---|---|---|---|---|---|---|---|---|---|---|
| | Median | Mean | SD | Minimum | Maximum | Median | Mean | SD | Minimum | Maximum | N | Per cent | N | Per cent |
| Energy (kcal) | 2126 | 2190 | 627 | 748 | 5085 | 2115 | 2179 | 626 | 748 | 5101 | 892 | 7.9 | 0 | 0 |
| Energy (kJ) | 8947 | 9222 | 2633 | 3124 | 21 394 | 8900 | 9175 | 2630 | 3124 | 21 440 | 891 | 7.9 | 0 | 0 |
| Protein (g) | 83.4 | 85.2 | 22 | 23.3 | 319.8 | 83.2 | 84.9 | 22 | 23.3 | 318.4 | 464 | 4.1 | 0 | 0 |
| Alcohol (g) | 6.7 | 12.3 | 16.1 | 0 | 134.2 | 6.7 | 12.3 | 16.1 | 0 | 134.2 | 0 | 0 | 0 | 0 |
| Carbohydrate (g) | 261 | 271 | 87 | 48 | 737 | 259 | 269 | 87 | 48 | 729 | 726 | 6.5 | 0 | 0 |
| Starch (g) | 123 | 128 | 45 | 10 | 504 | 122 | 127 | 45 | 10 | 501 | 813 | 7.2 | 1 | 0 |
| Englyst fibre (g) | 17.5 | 18.2 | 6.4 | 1.3 | 89.9 | 17.3 | 18 | 6.4 | 1.3 | 89.9 | 743 | 6.6 | 1 | 0 |
| Fat (g) | 78.9 | 83.2 | 31.3 | 13.4 | 260.6 | 78.7 | 83 | 31.3 | 13.4 | 260.6 | 1049 | 9.3 | 8 | 0.1 |
| Monounsaturated fat (g) | 27 | 28.8 | 11.6 | 4.8 | 101.2 | 26.8 | 28.5 | 11.5 | 4.8 | 105.1 | 1264 | 11.2 | 21 | 0.2 |
| Polyunsaturated fat (g) | 13.5 | 15 | 6.9 | 1.6 | 66.6 | 13.7 | 15.3 | 7.1 | 1.6 | 69.5 | 1074 | 9.5 | 24 | 0.2 |
| Saturated fat (g) | 30.1 | 32.3 | 13.6 | 3 | 110.6 | 29.8 | 31.9 | 13.5 | 3 | 106.7 | 1288 | 11.5 | 20 | 0.2 |
| Calcium (mg) | 1021 | 1039 | 301 | 189 | 2848 | 1018 | 1037 | 300 | 189 | 2849 | 296 | 2.6 | 1 | 0 |
| Iron (mg) | 12.1 | 12.4 | 3.6 | 2.6 | 38.7 | 11.9 | 12.3 | 3.5 | 2.5 | 38.5 | 1149 | 10.2 | 7 | 0.1 |
| Potassium (mg) | 3814 | 3881 | 911 | 1305 | 11 718 | 3802 | 3869 | 909 | 1284 | 11 718 | 411 | 3.7 | 0 | 0 |
| Carotene (μg) | 3188 | 3321 | 1573 | 147 | 25 720 | 3178 | 3309 | 1571 | 147 | 25 720 | 156 | 1.4 | 0 | 0 |
| Folate (μg) | 320 | 331 | 97 | 77 | 1547 | 316 | 327 | 96 | 77 | 1547 | 836 | 7.4 | 3 | 0 |
| Vitamin C (mg) | 103 | 111 | 52 | 10 | 669 | 105 | 113 | 52 | 10 | 669 | 411 | 3.7 | 14 | 0.1 |
| Vitamin D (μg) | 3.16 | 3.65 | 2.08 | 0.03 | 27.08 | 3.13 | 3.62 | 2.06 | 0.03 | 27.12 | 1161 | 10.3 | 145 | 1.3 |
| Vitamin E (mg) | 13.2 | 14.9 | 7.2 | 2.1 | 62.3 | 12.9 | 14.4 | 6.8 | 2.1 | 62 | 1545 | 13.7 | 146 | 1.3 |

CAFÉ, compositional analyses from frequency estimates; EPIC, European prospective investigation into cancer and nutrition; FETA, food frequency questionnaire EPIC tool for analysis.

**Table 2** Average daily nutrient intakes for women (N=13 602) participating in the EPIC-Norfolk study, from the FETA and CAFÉ programmes, after the exclusion of outliers, with numbers and percentages of women who moved quintile

| Nutrient | FETA programme | | | | | CAFÉ programme | | | | | Quintile change | | Quintile change >1 | |
|---|---|---|---|---|---|---|---|---|---|---|---|---|---|---|
| | Median | Mean | SD | Minimum | Maximum | Median | Mean | SD | Minimum | Maximum | N | Per cent | N | Per cent |
| Energy (kcal) | 1859 | 1925 | 546 | 538 | 4733 | 1853 | 1920 | 547 | 518 | 4643 | 1030 | 7.6 | 0 | 0 |
| Energy (kJ) | 7833 | 8113 | 2296 | 2261 | 19 910 | 7811 | 8091 | 2298 | 2179 | 19 537 | 1018 | 7.5 | 0 | 0 |
| Protein (g) | 79.8 | 81.5 | 21.1 | 23 | 246 | 79.6 | 81.3 | 21 | 22.7 | 246.1 | 495 | 3.6 | 1 | 0 |
| Alcohol (g) | 2 | 5.6 | 8.4 | 0 | 99.5 | 2 | 5.6 | 8.4 | 0 | 99.5 | 0 | 0 | 0 | 0 |
| Carbohydrate (g) | 237 | 247 | 77 | 59 | 766 | 235 | 245 | 77 | 58 | 766 | 974 | 7.2 | 1 | 0 |
| Starch (g) | 107 | 112 | 39 | 13 | 405 | 106 | 111 | 39 | 13 | 406 | 1142 | 8.4 | 1 | 0 |
| Englyst fibre (g) | 18.2 | 19 | 6.8 | 2.3 | 118.5 | 18 | 18.8 | 6.7 | 2.4 | 118.6 | 850 | 6.2 | 1 | 0 |
| Fat (g) | 67 | 70.8 | 27.1 | 11.7 | 221 | 67.2 | 71.2 | 27.3 | 11.6 | 217.2 | 1194 | 8.8 | 4 | 0 |
| Monounsaturated fat (g) | 22.5 | 24.1 | 9.9 | 3.8 | 100.3 | 22.5 | 24.1 | 9.9 | 3.5 | 100.6 | 1338 | 9.8 | 7 | 0.1 |
| Polyunsaturated fat (g) | 12.2 | 13.5 | 6.2 | 2 | 53.6 | 12.5 | 13.8 | 6.3 | 2 | 53.6 | 1434 | 10.5 | 23 | 0.2 |
| Saturated fat (g) | 25 | 27 | 11.7 | 3.6 | 102.3 | 25 | 26.9 | 11.7 | 3.7 | 99.3 | 1443 | 10.6 | 9 | 0.1 |
| Calcium (mg) | 971 | 992 | 290 | 128 | 3159 | 969 | 990 | 290 | 127 | 3159 | 390 | 2.9 | 4 | 0 |
| Iron (mg) | 11.5 | 11.8 | 3.6 | 1.7 | 66.1 | 11.3 | 11.7 | 3.5 | 1.8 | 65.7 | 1496 | 11 | 12 | 0.1 |
| Potassium (mg) | 3781 | 3861 | 942 | 1150 | 16 568 | 3769 | 3848 | 939 | 1147 | 16 587 | 486 | 3.6 | 1 | 0 |
| Carotene (μg) | 3477 | 3719 | 1917 | 67 | 61 971 | 3469 | 3712 | 1917 | 64 | 61 983 | 122 | 0.9 | 0 | 0 |
| Folate (μg) | 322 | 332 | 103 | 65 | 2039 | 317 | 328 | 101 | 65 | 2024 | 1025 | 7.5 | 5 | 0 |
| Vitamin C (mg) | 123 | 133 | 64 | 4 | 1006 | 125 | 135 | 64 | 4 | 1006 | 746 | 5.5 | 35 | 0.3 |
| Vitamin D (μg) | 3.01 | 3.46 | 1.9 | 0 | 17.83 | 3.02 | 3.45 | 1.9 | 0 | 17.75 | 1119 | 8.2 | 90 | 0.7 |
| Vitamin E (mg) | 12.4 | 13.8 | 6.2 | 1.5 | 52.4 | 12.2 | 13.5 | 6 | 1.6 | 49.8 | 1863 | 13.7 | 123 | 0.9 |

CAFÉ, compositional analyses from frequency estimates; EPIC, European prospective investigation into cancer and nutrition; FETA, food frequency questionnaire EPIC tool for analysis.

**Table 3** Average daily food group intakes for men (N=11 250) and women (N=13 602) participating in the EPIC-Norfolk study, from the FETA programme

| Food group | Men | | | | | Women | | | | |
|---|---|---|---|---|---|---|---|---|---|---|
| | Median | Mean | SD | Minimum | Maximum | Median | Mean | SD | Minimum | Maximum |
| Alcoholic beverages (g) | 101 | 204 | 315 | 0 | 2483 | 23 | 64 | 109 | 0 | 1728 |
| Cereals and cereal products (g) | 242 | 260 | 127 | 0 | 1456 | 215 | 231 | 110 | 0 | 1172 |
| Eggs and egg dishes (g) | 18 | 17 | 15 | 0 | 225 | 14 | 16 | 14 | 0 | 236 |
| Fats and oils (g) | 31 | 36 | 22 | 0 | 207 | 27 | 30 | 20 | 0 | 218 |
| Fish and fish products (g) | 32 | 37 | 26 | 0 | 362 | 32 | 38 | 26 | 0 | 309 |
| Fruit (g) | 179 | 212 | 164 | 0 | 2654 | 238 | 278 | 201 | 0 | 3742 |
| Meat and meat products (g) | 99 | 106 | 54 | 0 | 856 | 91 | 94 | 48 | 0 | 606 |
| Milk and milk products (g) | 407 | 420 | 182 | 0 | 1303 | 386 | 410 | 175 | 0 | 1560 |
| Non-alcoholic beverages (g) | 1157 | 1177 | 396 | 0 | 3707 | 1150 | 1165 | 403 | 0 | 4501 |
| Nuts and seeds (g) | 0 | 3 | 9 | 0 | 228 | 0 | 3 | 9 | 0 | 188 |
| Potatoes (g) | 125 | 122 | 69 | 0 | 1007 | 116 | 112 | 64 | 0 | 1506 |
| Soups and sauces (g) | 43 | 58 | 54 | 0 | 1004 | 43 | 57 | 53 | 0 | 1376 |
| Sugars (g) | 53 | 64 | 50 | 0 | 572 | 37 | 48 | 42 | 0 | 541 |
| Vegetables (g) | 236 | 255 | 123 | 0 | 2398 | 262 | 284 | 143 | 0 | 3539 |

EPIC, European prospective investigation into cancer and nutrition; FETA, food frequency questionnaire EPIC tool for analysis.

In women, (table 5), quintile changes (>15%) were also most evident in the same five nutrients. However, almost 21% of women also changed quintile for iron. Once again, the 'cereals and cereal products' food group was the only food group where there was any difference, with 40 women moving one quintile.

## DISCUSSION

FETA provides a new, freely available, stand-alone tool that can produce nutrient and food group intake values from data collected using the EPIC-Norfolk FFQ. It makes the EPIC-Norfolk FFQ readily accessible to end-users and enables them to process and analyse nutritional data. The data can either be entered into a spreadsheet, using the instructions provided, or by using the specifically developed Microsoft Access form-based entry tool. The Access entry tool allows easier entry without requiring knowledge of specific food codes. The software for FETA for Windows and Linux can be downloaded from the website, as can the Microsoft Access data entry utility (http://www.srl.cam.ac.uk/epic/epicffq/). Users are encouraged to register with EPIC-Norfolk, as this enables them to request assistance and support. The various types of output (with four levels of information) available should prove beneficial to researchers, especially those requiring more detailed information. There is an ongoing need for information on the intake of food groups. While the data from either output 3 or 4 could be used to generate more detailed food group data, we have treated food groups as another type of nutrient—a pseudonutrient. The FETA input/look-up files can be easily modified to create new groups, greatly adding to the flexibility of the system for analysing food group consumption, while requiring no spreadsheet or programming skills on the part of the analyst. A helpful feature of FETA is the log

file which documents errors relating to FFQ data and/or default food codes assigned.

FETA was designed and based on the extensively validated EPIC-Norfolk FFQ, originally developed in 1988, to assess the nutrient and food group intake of 40–79-year-olds, who completed the FFQ between 1993 and 1997. The food list and look-up lists of milks, breakfast cereals and fats reflect this time period and the study population, as do the default milk, cereal, baking fat and frying fat codes assigned. However, the programme was created in such a way that it can be customised for different study populations, easily enabled by the separation of the processing algorithm in the FETA programme implementation from the data model text files. It is possible to delete/add foods and/or FFQ lines, and modify portion sizes as desired for a study. Nutrient data may also be easily modified or added. It is also possible for FETA to be used with other questionnaires containing a different set of line items or different numbers of frequencies.

Comparisons were carried out for a number of selected nutrients obtained from FETA and the previously validated CAFÉ programme. These showed that the nutrient output from both programmes were generally similar. All differences (>0.1%) found from the comparison of detailed food/nutrient data at the individual level for 12 500 participants from FETA and the CAFÉ programmes can be explained by one or more of the following reasons: up to four cereal foods assigned by FETA, as compared to a maximum of two cereal foods assigned by CAFÉ; differences in default baking and frying fat codes assigned; correction for muesli portion size in cereal data; exclusion of porridge from cereal data (free text); default codes assigned for milk, cereals or fats to participants using FETA (where no food codes were assigned by CAFÉ programme); rounding error (only where percentage absolute differences were

**Table 4** Comparison of average daily nutrient and food group intakes for men (N=1159) participating in the EPIC-Norfolk study, from the FETA programme, with and without the application of text matching

| Nutrient/food group | FETA programme, with text matching | | | | | FETA programme, without text matching | | | | | Quintile change | | Quintile change >1 | |
|---|---|---|---|---|---|---|---|---|---|---|---|---|---|---|
| | Median | Mean | SD | Minimum | Maximum | Median | Mean | SD | Minimum | Maximum | N | Per cent | N | Per cent |
| Energy (kcal) | 2095 | 2176 | 678 | 658 | 7766 | 2091 | 2170 | 678 | 658 | 7787 | 28 | 2.4 | 0 | 0 |
| Energy (kJ) | 8822 | 9161 | 2848 | 2780 | 32 555 | 8804 | 9138 | 2850 | 2780 | 32 647 | 26 | 2.2 | 0 | 0 |
| Protein (g) | 82.8 | 85 | 22.8 | 22.1 | 272.3 | 82.5 | 84.7 | 22.8 | 22.1 | 272.3 | 34 | 2.9 | 0 | 0 |
| Alcohol (g) | 7.2 | 12.3 | 16.1 | 0 | 112.9 | 7.2 | 12.3 | 16.1 | 0 | 112.9 | 0 | 0 | 0 | 0 |
| Carbohydrate (g) | 261 | 270 | 93 | 63 | 1006 | 259 | 269 | 93 | 63 | 1003 | 48 | 4.1 | 0 | 0 |
| Starch (g) | 120 | 127 | 49 | 7 | 643 | 121 | 126 | 48 | 7 | 636 | 65 | 5.6 | 0 | 0 |
| Englyst fibre (g) | 17.5 | 18.3 | 6.6 | 3.6 | 71.8 | 17.3 | 17.9 | 6.3 | 3.6 | 64.5 | 198 | 17.1 | 10 | 0.9 |
| Fat (g) | 77.8 | 82.1 | 33.1 | 12.8 | 387.8 | 77.3 | 82.1 | 33.1 | 12.8 | 389.3 | 32 | 2.8 | 0 | 0 |
| Monounsaturated fat (g) | 26.5 | 28.2 | 12.2 | 3.5 | 131.1 | 26.7 | 28.7 | 12.5 | 3.7 | 138.7 | 88 | 7.6 | 0 | 0 |
| Polyunsaturated fat (g) | 13.5 | 14.9 | 7.3 | 3 | 67 | 12.7 | 14.1 | 6.8 | 3 | 60.7 | 179 | 15.4 | 17 | 1.5 |
| Saturated fat (g) | 30.1 | 31.8 | 14.1 | 3.3 | 160 | 30.3 | 32.2 | 14.3 | 3.3 | 160.3 | 72 | 6.2 | 1 | 0.1 |
| Calcium (mg) | 1015 | 1044 | 312 | 242 | 2848 | 1012 | 1044 | 313 | 242 | 2861 | 42 | 3.6 | 0 | 0 |
| Iron (mg) | 11.9 | 12.5 | 3.8 | 2.6 | 37.9 | 11.7 | 12 | 3.5 | 2.6 | 38.1 | 173 | 14.9 | 16 | 1.4 |
| Potassium (mg) | 3824 | 3889 | 957 | 1353 | 12 675 | 3812 | 3873 | 951 | 1353 | 12 551 | 52 | 4.5 | 0 | 0 |
| Carotene (μg) | 3150 | 3348 | 1671 | 507 | 18 295 | 3162 | 3353 | 1672 | 507 | 18 338 | 6 | 0.5 | 0 | 0 |
| Folate (μg) | 325 | 333 | 103 | 94 | 1222 | 316 | 326 | 101 | 94 | 1262 | 226 | 19.5 | 2 | 0.2 |
| Vitamin C (mg) | 105 | 113 | 55 | 17 | 619 | 104 | 112 | 55 | 17 | 619 | 22 | 1.9 | 0 | 0 |
| Vitamin D (μg) | 3.08 | 3.64 | 2.17 | 0.03 | 16.4 | 3.06 | 3.64 | 2.19 | 0.03 | 20.52 | 227 | 19.6 | 8 | 0.7 |
| Vitamin E (mg) | 13.3 | 15 | 7.6 | 2.7 | 74.7 | 13 | 14.5 | 7.1 | 2.7 | 71.2 | 238 | 20.5 | 30 | 2.6 |
| Alcoholic beverages (g) | 104 | 201 | 301 | 0 | 1866 | 104 | 201 | 301 | 0 | 1866 | 0 | 0 | 0 | 0 |
| Cereals and cereal products (g) | 240 | 257 | 131 | 0 | 1378 | 238 | 255 | 130 | 0 | 1378 | 31 | 2.7 | 0 | 0 |
| Eggs and egg dishes (g) | 18 | 17 | 17 | 0 | 225 | 18 | 17 | 17 | 0 | 225 | 0 | 0 | 0 | 0 |
| Fats and oils (g) | 31 | 36 | 25 | 0 | 313 | 31 | 36 | 25 | 0 | 313 | 0 | 0 | 0 | 0 |
| Fish and fish products (g) | 32 | 37 | 25 | 0 | 153 | 32 | 37 | 25 | 0 | 153 | 0 | 0 | 0 | 0 |
| Fruit (g) | 184 | 216 | 158 | 0 | 1037 | 184 | 216 | 158 | 0 | 1037 | 0 | 0 | 0 | 0 |
| Meat and meat products (g) | 98 | 104 | 52 | 0 | 690 | 98 | 104 | 52 | 0 | 690 | 0 | 0 | 0 | 0 |
| Milk and milk products (g) | 414 | 428 | 187 | 0 | 1302 | 414 | 428 | 187 | 0 | 1302 | 0 | 0 | 0 | 0 |
| Non-alcoholic beverages (g) | 1159 | 1191 | 397 | 22 | 3677 | 1159 | 1191 | 397 | 22 | 3677 | 0 | 0 | 0 | 0 |
| Nuts and seeds (g) | 0 | 3 | 8 | 0 | 135 | 0 | 3 | 8 | 0 | 135 | 0 | 0 | 0 | 0 |
| Potatoes (g) | 125 | 121 | 78 | 0 | 1518 | 125 | 121 | 78 | 0 | 1518 | 0 | 0 | 0 | 0 |
| Soups and sauces (g) | 43 | 56 | 51 | 0 | 556 | 43 | 56 | 51 | 0 | 556 | 0 | 0 | 0 | 0 |
| Sugars (g) | 51 | 63 | 50 | 0 | 358 | 51 | 63 | 50 | 0 | 358 | 0 | 0 | 0 | 0 |
| Vegetables (g) | 238 | 256 | 128 | 15 | 1047 | 238 | 256 | 128 | 15 | 1047 | 0 | 0 | 0 | 0 |

EPIC, European prospective investigation into cancer and nutrition; FETA, food frequency questionnaire EPIC tool for analysis.

**Table 5** Comparison of average daily nutrient and food group intakes for women (N=1340) participating in the EPIC-Norfolk study, from the FETA programme, with and without the application of text matching

| Nutrient/food group | FETA programme, with text matching | | | | | FETA programme, without text matching | | | | | Quintile change | | Quintile change >1 | |
|---|---|---|---|---|---|---|---|---|---|---|---|---|---|---|
| | Median | Mean | SD | Minimum | Maximum | Median | Mean | SD | Minimum | Maximum | N | Per cent | N | Per cent |
| Energy (kcal) | 1886 | 1946 | 607 | 608 | 8103 | 1880 | 1941 | 605 | 608 | 8134 | 50 | 3.7 | 0 | 0 |
| Energy (kJ) | 7938 | 8202 | 2554 | 2552 | 34 410 | 7909 | 8177 | 2547 | 2552 | 34 541 | 47 | 3.5 | 0 | 0 |
| Protein (g) | 80.3 | 82.5 | 22.2 | 26.8 | 277 | 79.9 | 82.1 | 22.1 | 26.8 | 276.6 | 43 | 3.2 | 0 | 0 |
| Alcohol (g) | 2 | 5.4 | 8.1 | 0 | 65.3 | 2 | 5.4 | 8.1 | 0 | 65.3 | 0 | 0 | 0 | 0 |
| Carbohydrate (g) | 238 | 250 | 90 | 67 | 1596 | 237 | 249 | 90 | 67 | 1603 | 58 | 4.3 | 0 | 0 |
| Starch (g) | 109 | 114 | 52 | 25 | 1288 | 108 | 114 | 52 | 25 | 1301 | 99 | 7.4 | 0 | 0 |
| Englyst fibre (g) | 18.6 | 19.3 | 7.4 | 4.1 | 103.7 | 17.8 | 18.7 | 7.1 | 3.3 | 97.2 | 247 | 18.4 | 13 | 1 |
| Fat (g) | 67.6 | 71.4 | 28.5 | 17.2 | 259.4 | 67.5 | 71.3 | 28.4 | 17.2 | 259.7 | 45 | 3.4 | 0 | 0 |
| Monounsaturated fat (g) | 22.7 | 24.4 | 10.6 | 4.8 | 104.2 | 23.1 | 24.6 | 10.6 | 4.8 | 103.8 | 133 | 9.9 | 0 | 0 |
| Polyunsaturated fat (g) | 12.2 | 13.6 | 6.2 | 2.6 | 42.5 | 11.5 | 12.9 | 5.9 | 2.5 | 39.4 | 224 | 16.7 | 11 | 0.8 |
| Saturated fat (g) | 25.2 | 27.2 | 12.4 | 5.1 | 109.6 | 25.5 | 27.5 | 12.4 | 5.1 | 109.6 | 74 | 5.5 | 2 | 0.1 |
| Calcium (mg) | 978 | 995 | 298 | 242 | 2528 | 976 | 992 | 297 | 242 | 2534 | 46 | 3.4 | 1 | 0.1 |
| Iron (mg) | 11.7 | 11.9 | 3.9 | 3.1 | 67.8 | 11.1 | 11.4 | 3.5 | 3.1 | 55.3 | 280 | 20.9 | 44 | 3.3 |
| Potassium (mg) | 3788 | 3874 | 994 | 1284 | 12 702 | 3744 | 3848 | 987 | 1280 | 12 526 | 68 | 5.1 | 0 | 0 |
| Carotene (μg) | 3489 | 3731 | 1705 | 178 | 13 796 | 3500 | 3736 | 1707 | 175 | 13 796 | 11 | 0.8 | 0 | 0 |
| Folate (μg) | 326 | 337 | 107 | 102 | 1311 | 318 | 329 | 105 | 97 | 1276 | 291 | 21.7 | 1 | 0.1 |
| Vitamin C (mg) | 124 | 133 | 63 | 4 | 809 | 122 | 132 | 62 | 4 | 809 | 34 | 2.5 | 0 | 0 |
| Vitamin D (μg) | 3.07 | 3.49 | 1.89 | 0.22 | 12.06 | 3.02 | 3.46 | 1.89 | 0.29 | 12.46 | 248 | 18.5 | 9 | 0.7 |
| Vitamin E (mg) | 12.5 | 13.8 | 6.3 | 2.7 | 52.4 | 12.1 | 13.3 | 5.9 | 3.3 | 43.6 | 270 | 20.2 | 21 | 1.6 |
| Alcoholic beverages (g) | 21 | 61 | 104 | 0 | 1350 | 21 | 61 | 104 | 0 | 1350 | 0 | 0 | 0 | 0 |
| Cereals and cereal products (g) | 214 | 236 | 174 | 9 | 4948 | 212 | 234 | 174 | 9 | 4948 | 40 | 3 | 0 | 0 |
| Eggs and egg dishes (g) | 14 | 16 | 14 | 0 | 136 | 14 | 16 | 14 | 0 | 136 | 0 | 0 | 0 | 0 |
| Fats and oils (g) | 27 | 30 | 19 | 0 | 133 | 27 | 30 | 19 | 0 | 133 | 0 | 0 | 0 | 0 |
| Fish and fish products (g) | 32 | 39 | 26 | 0 | 187 | 32 | 39 | 26 | 0 | 187 | 0 | 0 | 0 | 0 |
| Fruit (g) | 238 | 277 | 199 | 0 | 2830 | 238 | 277 | 199 | 0 | 2830 | 0 | 0 | 0 | 0 |
| Meat and meat products (g) | 90 | 95 | 49 | 0 | 392 | 90 | 95 | 49 | 0 | 392 | 0 | 0 | 0 | 0 |
| Milk and milk products (g) | 381 | 410 | 174 | 0 | 959 | 381 | 410 | 174 | 0 | 959 | 0 | 0 | 0 | 0 |
| Non-alcoholic beverages (g) | 1148 | 1153 | 404 | 8 | 3215 | 1148 | 1153 | 404 | 8 | 3215 | 0 | 0 | 0 | 0 |
| Nuts and seeds (g) | 0 | 3 | 11 | 0 | 180 | 0 | 3 | 11 | 0 | 180 | 0 | 0 | 0 | 0 |
| Potatoes (g) | 116 | 113 | 61 | 0 | 785 | 116 | 113 | 61 | 0 | 785 | 0 | 0 | 0 | 0 |
| Soups and sauces (g) | 45 | 57 | 53 | 0 | 900 | 45 | 57 | 53 | 0 | 900 | 0 | 0 | 0 | 0 |
| Sugars (g) | 38 | 50 | 46 | 0 | 540 | 38 | 50 | 46 | 0 | 540 | 0 | 0 | 0 | 0 |
| Vegetables (g) | 265 | 288 | 140 | 2 | 1387 | 265 | 288 | 140 | 2 | 1387 | 0 | 0 | 0 | 0 |

EPIC, European prospective investigation into cancer and nutrition; FETA, food frequency questionnaire EPIC tool for analysis.

between 0.1% and 1%) and changes made to the nutrient data of six of the nine new foods as well as to the default code for milk. A section entitled 'What are the differences between FETA versus CAFÉ processing?' found at http://www.srl.cam.ac.uk/epic/epicffq/FAQs.html further explains the aforementioned differences.

Although nutrient intakes as calculated by FETA and CAFÉ were similar, some relatively small differences existed, but these and the quintile shift of men and women can be explained. In FETA, a number of changes were made to the processing of breakfast cereals, affecting carbohydrate, starch, Englyst fibre, iron and folate estimates. The vitamin C content per 100 g of low-calorie/diet fizzy drinks was changed from 5 to 0 mg, and the vitamin E content of crunchy oat cereal and oil and fat non-specific was increased. Changes made to the processing of fats in questions 6 and 7 in part 2 of the FFQ, in addition to changes made to the fatty acid profile of the three new fats, could help explain the small differences observed in monounsaturated, polyunsaturated and saturated fat intakes.

There was quite a large range in intake in the 14 food groups, with a minimum intake of zero for each of the food groups. It is difficult to compare food group intake data as the groupings of foods often vary. However, the combined mean intake of fruit (excluding juices) and vegetables for men and women was 467 and 562 g respectively, achieving the Government's 'Five a day' recommendation,[23] using a portion size of 80 g.

While text matching only affected one food group (cereals and cereal products), more than 15% of men and women changed quintile for a number of nutrients: Englyst fibre, polyunsaturated fat, folate, vitamin D and vitamin E, and iron (women only). Yet again, these nutrients related to the text matching of breakfast cereals and baking and frying fats. The inclusion of these data illustrates the effect of text matching on the ranking of individuals for certain nutrients and will enable future researchers using FETA to make informed decisions on the benefit of text matching for their study.

We have not addressed or discussed common FFQ issues, such as the number of items in a food list or the use of a single average portion size, as these are not the focus of this paper and have been reviewed previously.[24 25]

It is anticipated that future updates of FETA might contain a number of improvements and overcome some of the limitations of FETA, currently released as V.2.53 for Windows and Linux (last updated 15 March and 21 February 2013, respectively). The source code has been made available online which enables users to make modifications and improvements to the programme. Currently, we have made available Windows and Linux versions and it is hoped that an OS X version will follow soon. We are currently working on a LibreOffice version of the Microsoft Access form-based entry tool.

In conclusion, we have created a new, open source, stand-alone, cross-platform FFQ processing tool, FETA,

to produce nutrient and food group data for researchers using the EPIC-Norfolk FFQ. The tool produces similar nutrient and food group values to the previously validated CAFÉ programme, but is more accessible. Although FETA was designed and based on the EPIC-Norfolk FFQ, the programme was created in such a way that it can be customised for different study populations. It is anticipated that the development and availability of FETA will be a useful addition to the field of nutritional epidemiology and dietary public health.

**Author affiliations**
[1]European Prospective Investigation into Cancer and Nutrition, Department of Public Health and Primary Care, Strangeways Research Laboratory, University of Cambridge, Cambridge, UK
[2]MRC Epidemiology Unit, Institute of Metabolic Science, Addenbrooke's Hospital, University of Cambridge, Cambridge, UK
[3]EPIC, Department of Gerontology, Addenbrooke's Hospital, School of Clinical Medicine, University of Cambridge, Cambridge, UK

**Acknowledgements** The authors would like to thank Mr Adam Dickinson, senior data manager at the MRC Epidemiology Unit, and his team members for their contribution to project management of FETA; Professor Nick Wareham, as EPIC-Norfolk study PI; and Mr Jamal Natour, as FETA software developer. The authors would also like to thank all the participants of the EPIC-Norfolk study and the EPIC-Norfolk staff for their help with this work.

**Collaborators** Adam Dickinson; Nick Wareham; and Jamal Natour.

**Contributors** AAM, AB and RNL contributed to the software development and assisted in statistical analyses. AAM drafted the manuscript. DJP-S wrote the step-based graphical wizard for running FETA. APK created the Microsoft Access form-based entry tool. All authors approved the final manuscript.

**Funding** This study was supported by programme grants from the MRC Population Health Sciences Research Network (PHSRN), Cancer Research UK (C864/A8257) and the Medical Research Council (G0401527 and G1000143). NGF was supported by the Medical Research Council (MC_UP_A100_1003); APK is funded by a Wellcome Trust Clinical Research Fellowship.

**Competing interests** None.

**Ethics approval** Norwich Local Research Ethics Committee.

**Provenance and peer review** Not commissioned; externally peer reviewed.

**Data sharing statement** EPIC-Norfolk has a wide range of collaborators. Contact details, publications and the process for collaborating and data requests can be found on the website (http://www.epic-norfolk.org.uk).

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
