## [Reviewer comments · BMJ Open]

Some articles will have been accepted based in part or entirely on reviews undertaken for other BMJ Group journals. These will be reproduced where possible.

ARTICLE DETAILS

TITLE (PROVISIONAL)	A new tool for converting food frequency questionnaire data into nutrient and food group values: FETA research methods and availability
AUTHORS	Mulligan, Angela; Luben, Robert; Bhaniani, Amit; Parry-Smith, David; O'Connor, Laura; Khawaja, Anthony; Forouhi, Nita; Khaw, KayTee

VERSION 1 - REVIEW

REVIEWER	Dr Yannan Jiang The University of Auckland New Zealand
REVIEW RETURNED	15-Feb-2014

GENERAL COMMENTS	In ABSTRACT, the Conclusions have additional information that is not presented in previous sections. For example, "FETA...has the advantages of being open source, cross-platform and complete with a data entry form directly compatible with the software." This is not obvious from the Objectives and/or Results. There are also differences in reported numbers between this paper and the cited references (#2 and #8), on those who completed the questionnaire and who presented in the analysis. Although this may not affect the overall conclusions, it should still be noted and if necessary discussed. The Statistical analyses section should give more details on any methods used for comparison, not only the summary statistics. Apart from the quintile changes used to compare the two programs, is there any other statistics/tests used to compare the nutrient and food group intake data as reported? If yes, they should be presented here to support those statements on "differences" and "similar/higher" etc. The Quintile changes reported in tables may be read better if the columns are separated from other summary statistics. The EPIC-Norfolk is a large cohort study with more than 25,000 participants who completed the questionnaire between 1993-1997. Since the FETA program can be customised for different study populations, is there any up-to-date data available on the population of interest?
--

REVIEWER	Amy F. Subar National Cancer Institute United States
REVIEW RETURNED	19-Feb-2014

GENERAL COMMENTS	All in all, this is a useful methodological paper that will be of use to the nutrition community. I only have a few comments that I think will clarify the text. 1. Line 84: I think you need to better explain the free-text issue rather than say it has been described elsewhere. A few examples here would help guide the reader... Or leave this out in the intro as it is better described in the methods. 2. Lines 121-125: I don't understand why these FFQs had to be manually entered -- were they not machine scannable? (It seems that is the case.) This is surprising to me but so be it. How did you insure the accuracy of manually entered data? Was there at least some double keying? It seems prone to error. 3. Lines 151- 156: Did the changes discussed here reflect intakes at the time the FFQ was administered or current nutrient intakes. Shouldn't it reflect intake at the time the FFQ was completed? 4. Line 191-194: Please list the number of exclusions for each criterion -- easier for the reader than having to do the math. 5. Lines 231 - 234: At this point, I was really wondering if the nutrient values applied were different because you used current vs baseline values. You say later that you used values applicable at baseline. It would have been easier for me if you had said this in the methods. 6. Lines 251-253 and 263 - 270: I think it is a good idea to make tools available to the public. What I don't understand in this specific instance is why someone would want to use this FFQ and its associated analytic tool if it has a database applicable to baseline. It is great to have a tool in which foods and portions can be modified. Can the nutrient database be modified as well? I don't see this stated. 7. Line 275: Why was the default backing and frying fat codes modified? I am not clear on the reason for all the changes you list through line 280. Can you say why you made changes to nutrient data for 6 of 9, etc? I am sure you have good reasons for this.
--

VERSION 1 – AUTHOR RESPONSE

Reviewer Name Dr Yannan Jiang

Institution and Country The University of Auckland

New Zealand

Please state any competing interests or state 'None declared': None declared

1) In ABSTRACT, the Conclusions have additional information that is not presented in previous sections. For example, "FETA...has the advantages of being open source, cross-platform and complete with a data entry form directly compatible with the software." This is not obvious from the Objectives and/or Results.

Line 35: The first sentence of the Objectives has been amended to "To describe the research methods for the development of a new open source, cross-platform tool which processes data from the European Prospective Investigation into Cancer and Nutrition Norfolk Food Frequency Questionnaire (EPIC-Norfolk FFQ)."

2) There are also differences in reported numbers between this paper and the cited references (#2 and #8), on those who completed the questionnaire and who presented in the analysis. Although this may not affect the overall conclusions, it should still be noted and if necessary discussed.

There are slight differences in the numbers quoted in this paper as compared to those in references 2 and 8. These differences result from data cleaning and the correction and exclusion of erroneous values in the years following the earlier publications. The overall data has not been affected by these differences.

3) The Statistical analyses section should give more details on any methods used for comparison, not only the summary statistics. Apart from the quintile changes used to compare the two programs, is there any other statistics/tests used to compare the nutrient and food group intake data as reported? If yes, they should be presented here to support those statements on "differences" and "similar/higher" etc.

No other statistics were used to compare the nutrient and food group intake data.

4) The Quintile changes reported in tables may be read better if the columns are separated from other summary statistics.

The space before the Quintile changes has been increased in the relevant tables.

5) The EPIC-Norfolk is a large cohort study with more than 25,000 participants who completed the questionnaire between 1993-1997. Since the FETA program can be customised for different study populations, is there any up-to-date data available on the population of interest?

At the moment, there is no up-to-date information available. However, data from the 2nd FFQ (approximately 3 years after 1st FFQ) is in the process of being written up. It is hoped that this will shortly be followed by ffq data from the 3rd health check (approximately 13 years after 1st FFQ).

Reviewer Name Amy F. Subar

Institution and Country National Cancer Institute

United States

Please state any competing interests or state 'None declared': None declared.

All in all, this is a useful methodological paper that will be of use to the nutrition community. I only have a few comments that I think will clarify the text.

1. Line 84: I think you need to better explain the free-text issue rather than say it has been described elsewhere. A few examples here would help guide the reader... Or leave this out in the intro as it is better described in the methods.

Lines 84-86: A further sentence has been added to try to better explain free-text matching: "Free text matching refers to the assigning of an appropriate food code to hand-written text in the FFQ and will be further described in the methods section".

This is more fully described in the methods section but felt it needed to be touched upon in the introduction as its effect is one of the aims of this paper.

2. Lines 121-125: I don't understand why these FFQs had to be manually entered -- were they not machine scannable? (It seems that is the case.) This is surprising to me but so be it. How did you insure the accuracy of manually entered data? Was there at least some double keying? It seems prone to error.

The questionnaires were not designed to be machine scannable, and included several sections where a written response was needed and scanning would not have been possible. While double data entry was not used, subsequent checks on the data suggested a very low error rate and invalid data items have been changed or excluded.

3. Lines 151- 156: Did the changes discussed here reflect intakes at the time the FFQ was administered or current nutrient intakes. Shouldn't it reflect intake at the time the FFQ was completed?

Lines 154-160: The changes made here reflected the nutrient content of foods at the time the FFQs were administered. This has now been clarified in the document: "Modifications to the nutrient data were made to ensure a more accurate nutrient profile and/or to better reflect the foods consumed, in the case of non-specific items, such as milk and oil/fat; these changes relate to nutrient/food data at the time of FFQ completion."

4. Line 191-194: Please list the number of exclusions for each criterion -- easier for the reader than having to do the math.

Lines 195-202: This section has been revised and the relevant numbers have been inserted: "We received FFQs from 25 351 participants (11 451 men and 13 900 women), with a mean age of 59 years. From this set, 249 FFQs (90 men and 159 women) containing 10 or more missing lines of data in Part 1 of the FFQ were excluded, followed by a further exclusion of 250 FFQs (111 men and 139 women) from the top and bottom 0.5% of EI:BMR. This resulted in the final analytical dataset of 24 852 participants (11 250 men and 13 602 women)."

5. Lines 231 - 234: At this point, I was really wondering if the nutrient values applied were different because you used current vs baseline values. You say later that you used values applicable at baseline. It would have been easier for me if you had said this in the methods.

Lines 157-160: The nutrient values and foods used relate to the time that the FFQs were completed. This has now been clearly stated in the methods: "Modifications to the nutrient data were made to ensure a more accurate nutrient profile and/or to better reflect the foods consumed, in the case of non-specific items, such as milk and oil/fat; these changes relate to nutrient/food data at the time of FFQ completion."

6. Lines 251-253 and 263 - 270: I think it is a good idea to make tools available to the public. What I don't understand in this specific instance is why someone would want to use this FFQ and its associated analytic tool if it has a database applicable to baseline. It is great to have a tool in which foods and portions can be modified. Can the nutrient database be modified as well? I don't see this stated.

The EPIC-Norfolk FFQ is widely used and has been extensively validated and so was chosen as the FFQ to use when creating FETA. Although FETA is based on this FFQ, the system was set up in such a way that it is easy to amend to suit different FFQ study populations. It is possible to add/delete foods, amend portions and add/amend nutrient data. It is also possible for FETA to be used with other questionnaires containing a different set of line items or different numbers of frequencies.

Researchers are encouraged to register with us and in turn, will receive advice and assistance as required.

Lines 277-278: Also, two new sentences have been added: "Nutrient data may also be easily modified or added. It is also possible for FETA to be used with other questionnaires containing a different set of line items or different numbers of frequencies."

7. Line 275: Why was the default backing and frying fat codes modified? I am not clear on the reason for all the changes you list through line 280. Can you say why you made changes to nutrient data for 6 of 9, etc? I am sure you have good reasons for this.

Lines 290-292: A sentence has been added to the manuscript with a link providing information on the main differences between CAFÉ and FETA processing: "A section entitled 'What are the differences between FETA versus CAFÉ processing?' found at <http://www.srl.cam.ac.uk/epic/epicffq/FAQs.html> further explains the aforementioned differences. "

Please see below for its content:

Breakfast cereals

In contrast to CAFÉ, which was only able to deal with a maximum of two breakfast cereals, FETA allows up to four cereal types to be recorded and portion weights are adjusted accordingly. Of the 24 633 participants who consumed breakfast cereal, 22 508 consumed either one or two cereals whereas 2 125 consumed either three or four breakfast cereals. The average portion size for all breakfast cereals is 30g, with the exception of muesli, which is 60g.

Some participants recorded porridge as one of their breakfast cereals. However, porridge consumption should be quantified in line 23 in Part 1 of the FFQ (see Figure 1). In the CAFÉ program, porridge recorded in Part 2 was processed, but the FETA program excludes any food codes relating to 'porridge' type items from the cereal look-up list.

Frying and baking fats

Occasionally, the free text entered for vegetable oil in Question 6 is the same as that entered for margarine in Question 7. In the CAFÉ system, only one food code could be assigned to unique free text. Therefore, if it was decided that 'sunflower' meant sunflower oil, (Question 6), when 'sunflower' was noted for the type of margarine used in baking, (Question 7), it would also have the food code of sunflower oil assigned to it. However, in the spreadsheet for FETA entry, improved layout and data entry ensures that the most appropriate food code is assigned.

Selection of the 'none' or 'No' box and default milk, cereal, and fat codes

Sometimes, the 'None' or 'No' box in part 2 is ticked but further information provided and/or assumptions made, result in the assigning of a default code. For example, if an individual records that they consume milk in Question 4, but tick the 'None' box in Question 3, a default milk code is assigned.

A number of EPIC-Norfolk participants ticked the 'none' box for baking (N=3 925) and frying (N=903) fats, though it was thought more likely that fat was used but the type was unknown. In FETA, if the 'none' box is ticked, the appropriate default fat is used in the absence of free text, while in the CAFÉ program, 'none' was assumed to mean no fat used.

The default baking and frying fats are taken from the Miscellaneous Foods supplement (19); these food codes used in the CAFÉ program have been changed in FETA to more appropriate codes. The default milk was calculated using 50% semi-skimmed milk, 25% whole milk and 25% skimmed milk (the default mapping used in CAFÉ was 50 % whole milk, 40% semi-skimmed milk and 10% skimmed milk).

These default codes are also applied by the program, as required, when specific food codes can not be assigned by text matching.